# Ancient Horizontal Gene Transfers from Plastome to Mitogenome of a Nonphotosynthetic Orchid, *Gastrodia pubilabiata* (Epidendroideae, Orchidaceae)

**DOI:** 10.3390/ijms241411448

**Published:** 2023-07-14

**Authors:** Young-Kee Kim, Sangjin Jo, Se-Hwan Cheon, Ja-Ram Hong, Ki-Joong Kim

**Affiliations:** 1Division of Life Sciences, Korea University, Seoul 02841, Republic of Korea; gimyoung2@korea.ac.kr (Y.-K.K.); sjjo0925@kribb.re.kr (S.J.); cheonsh@korea.ac.kr (S.-H.C.); ghdwkfka@korea.ac.kr (J.-R.H.); 2International Biological Material Research Center, Korea Research Institute of Bioscience and Biotechnology, Daejeon 34141, Republic of Korea

**Keywords:** Orchidaceae, plastome, mitogenome, horizontal gene transfer, mycoheterotorphy

## Abstract

*Gastrodia pubilabiata* is a nonphotosynthetic and mycoheterotrophic orchid belonging to subfamily Epidendroideae. Compared to other typical angiosperm species, the plastome of *G*. *pubilabiata* is dramatically reduced in size to only 30,698 base pairs (bp). This reduction has led to the loss of most photosynthesis-related genes and some housekeeping genes in the plastome, which now only contains 19 protein coding genes, three tRNAs, and three rRNAs. In contrast, the typical orchid species contains 79 protein coding genes, 30 tRNAs, and four rRNAs. This study decoded the entire mitogenome of *G*. *pubilabiata*, which consisted of 44 contigs with a total length of 867,349 bp. Its mitogenome contained 38 protein coding genes, nine tRNAs, and three rRNAs. The gene content of *G. pubilabiata* mitogenome is similar to the typical plant mitogenomes even though the mitogenome size is twice as large as the typical ones. To determine possible gene transfer events between the plastome and the mitogenome individual BLASTN searches were conducted, using all available orchid plastome sequences and flowering plant mitogenome sequences. Plastid rRNA fragments were found at a high frequency in the mitogenome. Seven plastid protein coding gene fragments (*ndh*C, *ndh*J, *ndh*K, *psa*A, *psb*F, *rpo*B, and *rps*4) were also identified in the mitogenome of *G*. *pubilabiata*. Phylogenetic trees using these seven plastid protein coding gene fragments suggested that horizontal gene transfer (HGT) from plastome to mitogenome occurred before losses of photosynthesis related genes, leading to the lineage of *G*. *pubilabiata*. Compared to species phylogeny of the lineage of orchid, it was estimated that HGT might have occurred approximately 30 million years ago.

## 1. Introduction

The advancement in sequencing technology has led to a surge in research on nucleotide sequences of flowering plants and their use for evolutionary and phylogenetic studies [1,2]. However, these studies have mainly focused on the plastome due to the cost and time constraints in sequencing. As of February 2023, 7588 species of angiosperm plastome sequences have been deposited in the NCBI, whereas only 347 mitogenome sequences have been deposited in the NCBI. Research using plastome sequences has been conducted on various flowering plants, such as *Amborella* [3], magnolids [4], monocots [5], and rosid [6]. Despite challenges in sequencing, nuclear genomes of some flowering plants have been decoded, including *Amborella* [7], *Oryza* [8], *Phalaenopsis* [9], and *Triticum* [10].

Both the mitochondria and chloroplast are essential organelles in plant function. The plastome is the most widely used genome in plant evolutionary study because it has relatively stable genome structure, size, and rich evolutionary information. In contrast to plastome, the mitogenome vary widely in size and structure across taxa even among different populations of same species [11,12]. The smallest mitogenome is 66 kb for *Viscum* [13], while the longest mitogenome is 11.3 mb for *Silene* [14]. It has been suggested that these differences result from genomic rearrangements, repetitive sequences, and/or foreign DNA insertions [15,16,17]. Due to many repeats, long length, and rearrangement of the mitogenome, it is difficult to decipher the overall structure of the mitogenome in majority of plant species. Frequent gene transfers have been reported in the mitogenome of *Valeriana* of photosynthetic Dipsacales [18]. Intracellular gene transfer (IGT) and horizontal gene transfer (HGT) have been reported in the parasitic plant *Aeginetia* [19]. Gene transfer has been reported during the symbiotic relationship between fungi and orchids, which are closely related to saprophytic lifestyles of the Orchidaceae [20]. The trace of gene transfer has been studied in a few orchid mitochondria. These imported sequences might be due to past biological contact [12,20,21,22].

The Orchidaceae are one of the largest families of flowering plants, containing approximately 25,000 species [23]. Recent studies have divided this family into five subfamilies: Apostasioideae, Vanilloideae, Orchidoideae, Cypripedioideae, and Epidendroideae [24]. Following the first complete plastome sequencing report of *Phalaenopsis aphrodite* [25], plastome sequencing reports have been published for all major lineages within Orchidaceae, such as *Corallorhiza* [26,27], *Cymbidium* [28], *Dendrobium* [29], *Holcoglossum* [30], *Neottia* [31], and *Vanilla* [32]. To date, at least 321 orchid plastome sequences have been deposited in the NCBI (as of February 2023). The growing number of reported orchid plastome sequences has led to various studies of the Orchidaceae family, focusing on phylogenetics, evolution, and plastome structure [5,32,33,34,35,36,37,38]. These studies often concentrate on specific lineages such as epiphytic or non-photosynthetic orchids. There are a total of 43 genera that contain non-photosynthetic mycoheterotrophs [39]. Several plastome sequences of these types of orchids have been reported, such as *Aphyllorchis* [31], *Corallorhiza* [40], *Cyrtosia* [35], *Epipogium* [41], *Gastrodia* [42], *Hexalectris* [37], *Lecanorchis* [43] and *Rhizantella* [33]. Typical photosynthetic orchid plastomes contain 83 protein coding genes [25]. However, non-photosynthetic species such as *Epipogium roseum* have significantly reduced plastomes, containing only 17 protein coding genes [41]. This contraction of plastomes results in low levels of phylogenetically informative sites. Genes retained in these plastomes often exhibit elevated evolutionary rates [44], making it difficult to construct accurate phylogenetic trees and interpret their results [5,43].

For the Orchidaceae family, the mitogenome sequence has only been reported for *Gastrodia elata* [42] and nuclear genome sequences have only been reported for *Cymbidium* [45], *Dendrobium* [46], *Gastrodia* [42], and *Phalaenopsis* [9]. Despite active research into phylogenetic and evolutionary aspects of Orchidaceae, phylogenetic trees from different organelles are often inconsistent [43,47,48]. This might be due to factors such as incomplete lineage sorting, ancient plastome capture, and/or sampling issues [43]. To address these incongruences, more mitochondrial and nuclear genome sequence data are needed.

The species *Gastrodia pubilabiata* belongs to the tribe Gastrodieae of the subfamily Epidendroideae of the family Orchidaceae [24]. It is widely distributed in Korea, Japan, and Taiwan [49,50], but it is a rare and endangered species. For its closely related species, *G*. *elata*, plastome sequences along with nuclear and mitogenome sequences have been reported [42,43]. The plastome of *G*. *elata* is greatly reduced compared to that of a typical photosynthetic orchid. Flowers and stems of *G*. *pubilabiata* are pale greenish in color. In a previous study on *Cymbidium macrorhizon* [51] which also has pale greenish stems and flowers, the plastome was almost intact condition, although it is known as a non-photosynthetic species. Only a few *ndh* genes were missing from its plastome. Therefore, the extreme reduction in plastome sequence in *G*. *elata* raises the question of what type of plastome sequence will be present in *G*. *pubilabiata* which has pale greenish stems and flowers.

The current state of research on the plastome and mitogenome of non-photosynthetic orchids presents several difficulties, including a lack of parsimonious informative sites in the plastome and incongruence in phylogenetic trees based on different genomes. To address these issues, this study aimed to decode both of the plastome and mitogenome of the non-photosynthetic orchid *G*. *pubilabiata*. Through this research, organelle genomes of non-photosynthetic orchids will be better understood and IGT or HGT between different organelle genomes will be documented if it existed. In addition, any potential foreign nucleotide sequences in organelle genomes will be identified if it presented. Through this study, if any evidence of externally imported nucleotide sequences is found, the source of sequences will be investigated and the relationship with the taxon will be determined. Results of this study will contribute to the further understanding of the evolutionary modes of mitogenome in non-photosynthetic orchid species.

## 2. Results

### 2.1. Organelle Characteristics and Comparative Analysis

The plastome sequence of *G. pubilabiata* was a 30,698 bp in length (Table 1 and Figure 1). The GC content of the plastome was 24.9% with a depth coverage of 236.0×. Unlike other reported orchid plastome sequences such as *Cypripedium* or *Habenaria*, the plastome of *G*. *pubilabiata* was much shorter in length. In addition, it lacked the typical quadripartite structure of angiosperm plastomes, which typically consist of a large single copy (LSC) and a small single copy (SSC) separated by two inverted repeats (IRs). Instead, the plastome of *G*. *pubilabiata*, like that of *G*. *elata*, lacked IRs. As a non-photosynthetic, mycoheterotrophic orchid without any leaves, *G*. *pubilabiata* lost all photosynthetic related genes, including *ndh*, *pet*, *psa*, *psb*, and *rpo* as well as several housekeeping genes, *atp* genes, *rrn*5, *ccs*A, *cem*A, *inf*A, *mat*K, *rbc*L, *ycf*3, and *ycf*4. Finally, the plastome of *G*. *pubilabiata* was annotated. It had only 19 protein coding genes, three tRNA genes, and three rRNA genes.

As a result of de novo assembly, 44 mitochondrial contigs of *G. pubilabiata* were obtained (Table 1). The goal was to obtain a complete, master circle of the mitogenome of *G*. *pubilabiata*. However, it was proved to be challenging due to the presence of AT-rich regions, rearrangements, and the recombination. These 44 mitochondrial contigs were considered as the complete mitogenome with a total length of, 867,349 bp (Table 1 and Figure 2). The coverage depth was 150.0×, with the highest coverage being 627.2× (14,604 bp) and the lowest coverage being 40.1× (4510 bp). The GC content of the mitogenome was 42.1%. With these 44 mitochondrial contigs, a total of, 38 protein coding genes, nine tRNA genes, and three rRNA genes were annotated. Compared to other monocot mitogenomes, three genes (*rpl*10, *sdh*3, and *sdh*4), *nad*1 intron1, and *nad*1 intron2 were missing in the mitogenome of *G*. *pubilabiata* (Figure 2).

Repeat analysis was conducted to compare the mitogenome and plastome of *G. pubilabiata*. Three additional organelles were also included (the mitogenome of *G*. *elata*, the plastome of *G*. *elata*, and the plastome of *G*. *longistyla*) in this analysis. The analysis revealed a total of 437 repeats (Appendix A). Of these, 279 repeats were found in the mitogenome of *G*. *pubilabiata*, 79 repeats were shared between the mitogenomes of *G*. *pubilabiata* and *G*. *elata*, 44 repeats were shared among the three *Gastrodia* plastomes, 34 repeats were identified within the mitogenome of *G*. *elata*, and one repeat was shared between the *G*. *elata* plastome and mitogenome (Figure 3 and Appendix A). Additionally, 89 simple sequences repeats (SSRs) were found within the plastome of *G*. *pubilabiata*, including 16 mononucleotide SSRs, five dinucleotide SSRs, four trinucleotide SSRs, 13 tetranucleotide SSRs, and 51 pentanucleotide SSRs (Appendix A).

### 2.2. Foreign Sequences in the Organelle

BLASTN and BLASTP searches were performed for the plastome of *G. pubilabiata* to identify exotic regions. Results of the BLASTN showed that most regions had the highest matches with plastome sequences of green plants. However, there were also matches with algae, bacteria, and animal sequences in regions corresponding to rRNAs (*rrn*4.5, *rrn*5, *rrn*18, and *rrn*23). Results of the BLASTP for open reading frames (ORFs) in the *G*. *pubilabiata* plastome confirmed that most ORFs were of green plant origin. In some cases, ORFs located near or within *rpl*16, *rrn*16, *rrn*23, and *rps*12 showed a high proportion of matches with bacterial sequences in BLASTP search results.

During annotation of protein coding genes in the plastome of *G*. *pubilabiata*, no debris from other mitogenome or nuclear genomes was found. However, during annotation of mitogenome, sequences like plastome derived *psb*F genes were detected, which was 119 bp in length. Regarding this regions, similar regions were found in other previously reported mitogenomes. These sequences were aligned and used to construct a maximum likelihood tree (Appendix A). In this ML tree, all mitochondrial *psb*F-like regions formed a monophyletic group, indicating that mitochondrial and plastome *psb*F genes evolved independently.

The plastome and mitogenome of *G. pubilabiata* were searched using local BLASTN against two databases of published angiosperm plastomes and mitogenomes. As a result, 23 blocks of the *G*. *pubilabiata* plastome were found to match with the mitogenome database. Of these 23 blocks, 17 were related to *rrn*16 and *rrn*23, while the remaining were related to *clp*P, *rpl*2, *rpl*16, and *ycf*1. In the *G*. *pubilabiata* mitogenome, 118 blocks were identified as having plastome-related sequences, 82 of which had a higher degree of similarity to mitogenomes than to the plastome database. Thirty-six blocks were found to be plastome-related blocks, largely located in intergenic spacers or rRNAs. Six regions were found to be coding regions in the plastome (*ndh*C, *ndh*J, *ndh*K, *psa*A, *rpo*B, and *rps*4) (Appendix A). A phylogenetic tree was constructed using these regions after performing MAFFT alignment (Figure 4).

### 2.3. Phylogenetic Position of G. pubilabiata

The phylogenetic position of *G. pubilabiata* was determined using 79 protein coding regions and four rRNAs in its plastome. Results showed that *G*. *pubilabiata*, *G*. *elata*, and *G*. *longystyla* formed a monophyletic group. All nodes in the *Gastrodia* clade were supported by 100% bootstrap values (Figure 5). General phylogenetic position of the Orchidaceae was found to be Apostasioideae [Vanilloideae [Cypripedioideae [Orchidoideae, Epidendroideae]]].

As previously mentioned, studies decoding mitogenomes of orchids are limited. Thus, a phylogenetic tree was constructed using previously reported angiosperm mitogenomes. Results showed that *G*. *pubilabiata* and *G*. *elata* formed a monophyletic group and the genus *Gastrodia* formed a sister relationship with *Allium cepa* and *Asparagus officinalis*. Most species were supported by high bootstrap values (above 90%). However, species such as *Beta*, *Daucus*, *Fagus*, *Geranium*, *Luffa*, *Nelumbo*, and *Prunus* showed relatively lower bootstrap values. The phylogenetic tree based on the mitogenomes was in line with previously reported phylogenetic trees based on plastomes or nuclear genomes (Figure 6).

## 3. Discussion

### 3.1. Organelle Genome Evolution

The plastome of *G. pubilabiata* had a distinct feature in that it has extremely reduced genome size and consists of only a single copy region. This same plastome structure has previously been also reported in the same genus, *G*. *elata* [42]. Other taxa without IR include Orobanchaceae [52], Cactaceae [53], Geraniaceae [54,55], Fabaceae [56], and Gymnosperms [57,58]. In Fabaceae, a representative group of taxa lacking IR, deletions of genes or introns, rearrangements of the plastome structure, gene transfer, and hypermutations have been reported [59]. The lack of IR in the legume plastome might be a result of plastome rearrangement mediated by repetitive DNA [60]. These repeats are thought to have facilitated rearrangements by enabling recombination [55].

The plastome of *G. pubilabiata* contains simple sequence repeats (SSRs) located around *rps*2-*rps*14, *rps*2-*rps*3, and *ycf*2 regions (Appendix A). The gene *trn*Q-UUG is located between *rps*2 and *rps*14, and the *trn*C-GCA gene is situated near *rps*2 and *rps*3. Recombination of repeats and tRNAs might be the primary mechanism behind genomic rearrangements in *Trachelium* [61]. In a typical orchid plastome, *rps*2 is found near the LSC-IRa junction and *rps*3 is located near the LSC-IRb junction. *Rps*2 and *rps*14 are 20 kb apart. The change in position of *rps*2, *rps*3, and *rps*14 in *G*. *pubilabiata* is believed to be due to the presence of repeats or tRNAs in the intergenic spacer (IGS) near the relevant region. However, it is uncertain whether gene reduction, in which protein coding regions between corresponding genes are lost, precedes genome rearrangement and reduction [62]. Some studies have suggested that repetitive sequences contribute to genome rearrangement in the Geraniaceae [55]. Accumulation of SSRs and repeats in the plastome of *G*. *pubilabiata* might have also contributed to gene loss or plastome size reduction.

Several changes such as *ndh* gene contents, IR boundary shifts, and IR absence have been reported in plastomes of various orchid taxa [5,32,42,43,62]. For non-photosynthetic taxa, changes have also been reported in gene groups other than just *ndh* genes [43]. This level of change is consistent with previous reports for non-photosynthetic flowering plants [63]. In the case of *G*. *pubilabiata*, most genes have disappeared. The plastome of *G*. *pubilabiata* belongs to the fourth stage of plastome degradation pattern (Appendix A). Other orchid taxa that belong to stages 4 of plastome degradation, including *Epipogium* [41], *Gastrodia* [42], *Rhizanthella* [33], and *Lecanorchis* [43], are also non-photosynthetic mycoheterotrophs with greatly altered plastomes. These taxa have only retained some housekeeping genes.

Flowers and stems of *G. pubilabiata* have a green color, which sets them apart from the other fourth stages of orchid with red, white, or yellow flowers and stems. In contrast to the other green stemmed mycoheterotrophs *C. macrorhizon*, which only lacks *ndh* genes in its plastome [51], all photosynthesis related genes have disappeared from the confirmed plastome of *G*. *pubilabiata* (Figure 1 and Appendix A). Despite this, low level photosynthesis has been observed in flowers and stems of *C. macrorhizon* due to residual chlorophyll [64,65,66]. Given this and the presence of residual plastids in other non-photosynthetic taxa, it is hypothesized that there might be an alternative pathway for photosynthesis in *G*. *pubilabiata*. Further research such as RNA-seq is needed to confirm this.

Mitogenomes of flowering plants are known to be variable in structure due to the presence of repeats, rearrangements, and submolecules [67]. Mitogenomes of monocots can vary greatly in length and genetic composition (Table 2 and Appendix A). In the genus *Gastrodia*, multiple mitogenomes have been reported [42]. The mitogenome of *G*. *pubilabiata* was found to have 44 mitogenomes (Table 1 and Figure 2). Unlike the plastome, which displays a significant difference in genetic composition between photosynthetic and non-photosynthetic taxa, it is believed that gene composition of the mitogenome is not significantly impacted by photosynthesis or the lack thereof.

### 3.2. Putative Gene Transfer

BLAST search was performed to detect any foreign sequences in the plastome and mitogenome of *G. pubilabiata*. The search uncovered regions with high plastome frequency having mitochondrial features and regions with high mitochondrial frequency having plastome characteristics (Figure 7). Regions with high frequency of mitogenome even in the plastome were mostly made up of rRNA regions (Appendix A). Other protein coding regions in the plastome (*acc*D, *clp*P, *rpl*16, and *rpl*2) did show BLASTN results for the mitogenome. However, it was confirmed that there were many more search results for the plastome (Appendix A). The plastome might have infiltrated to the mitogenome, or there was a coincidental similarity of nucleotide sequences, rather than the presence of corresponding regions derived from the mitogenome. Reports of the presence of mitochondrial plastome sequences (MTPTs) have been documented in flowering plants [15]. A recent study has also reported the presence of plastome originating sequences in the mitogenome of *Mangifera* species [68].

BLASTN results for the mitogenome were like those for the plastome. Many results show the high frequency for the plastome contained regions having rRNA, such as 16 s rRNA, and 18 s rRNA, suggesting that there might have been an exchange between rRNAs of the mitogenome and the plastome. However, for some regions, BLASTN results were found for plastome regions (*ndh*C, *ndh*J, *ndh*K, *psa*A, *psb*F, *rpo*B, and *rps*4), but not for mitogenome references. This suggests that gene transfer from the plastome to the mitogenome might have occurred. The mitogenome contains many photosynthesis-related genes from the plastome (*ndh*, *psb*, *psa, rpo*), even though *G*. *pubilabiata* is a non-photosynthetic plant and all photosynthesis related genes have been deleted from the current plastome (Figure 1 and Appendix A). This suggests that transfer of photosynthesis related genes to the mitogenome might have occurred before the loss of photosynthetic function in plastome. Therefore, this gene transfer might have occurred a longer time ago than previously expected. Except for a few obligate mycoheterotrophic taxa, a previous study has shown that the divergence of mycoheterotrophs might have occurred earlier than 30 mya [43]. This suggests that photosynthesis-related genes might have disappeared after that time. Gene transfer might have occurred a longer than 30 mya. However, as pointed out in that previous study, research results on the plastome and mitogenome of related obligated mycoheterotrophic taxa will be needed to verify that hypothesis.

A phylogenetic tree was constructed to determine the origin of gene fragments (*ndh*, *psa*, *psb*, and *rpo*) in the mitogenome of *G*. *pubilabiata* with other major flowering plants. Results are shown in Figure 4. In the case of *psb*F, *G*. *pubilabiata* formed a monophyletic relationship with other mitochondrial *psb*F. It tended to be distinguished between the plastome and the mitogenome. This suggests that these gene fragments might have been transferred from the plastome to the mitogenome in the distant past. In the case of *psa*A, the mitogenome derived *psa*A displayed a long branch, which was believed to result from the absence of other mitogenome derived sequences with the same cause as the *psb*F phylogenetic tree. The phylogenetic tree using *rpo*B region was distinct from trees using *psa*A and *psb*F gene fragments. It did not show a long branch and *rpo*B sequences of *G*. *pubilabiata* from a monophyletic branch with *Euonymus* of Celastrales, which was supported by a bootstrap value of 100. This suggests that *G*. *pubilabiata* has a close relationship with Celastraceae plants in its vicinity. Similar relationships have been reported for parasitic plants such as *Aeginetia* with nearby Poaceae plants [19]. Although it is difficult to directly compare mycoheterotrophic orchid to parasitic plants, this orchid might be related to surrounding plants. It was confirmed that *Celastrus* and *Euonymus* lived some distance away, not nearby.

Unlike photosynthesis-related genes found in the mitogenome, the *rps*4 gene is classified as a housekeeping gene. The presence of this gene in both the plastome and mitogenome of *G*. *pubilabiata* was confirmed and a phylogenetic tree was constructed to examine its origin. The phylogenetic tree showed that both plastome and mitogenome derived sequences of the *rps*4 gene formed a monophyly with other orchids. However, the plastome derived sequence showed a long branch and formed a monophyly, while the mitogenome derived sequence was at the base of the clade. This difference in evolutionary rates between the two sequences was likely due to relaxed selection in degraded plastomes of mycoheterotrophic plants, which tended to drive the acceleration of gene evolution [69]. Moreover, mitochondrial datasets show much slower evolutionary rates than the plastome datasets in the previous study [70]. Thus, it could be concluded that the *rps*4 gene in the plastome evolved at a faster evolutionary rate than the gene in the mitogenome.

It is known that *ndh* genes are typically the first to be lost in the plastome degradation process of non-photosynthetic species, followed by other photosynthesis related genes [52]. This study found that *ndh*C, J, and K genes in the mitogenome of *G*. *pubilabiata* formed a monophyly with other orchids, unlike photosynthesis-related gene *rpo*, *psa*, or *psb*, which showed relationships with other taxa of flowering plants (Figure 4). This suggested that the degradation of *G*. *pubilabiata*’s plastome was more recent or that alternative pathways were involved. Reference guided assembly analysis only found a few base sequences for *ndh*C, J, and K in the mitogenome. No base sequence was found to support the possibility of transfer to the nuclear genome. Hence, it was assumed that *ndh*C, J, K genes disappeared more recently from the plastome or that the degradation of the plastome was more recent. While it was believed that non-photosynthesis in Orchidaceae developed within the past 30 mya [43], further experiments such as transcriptome analysis are needed to explore the possibility of alternative pathways.

### 3.3. Phylogenetic Implications

Phylogenetic relationships within the Orchidaceae are presented as Apostasioideae [Vanilloideae [Cypripedioideae [Orchidoideae, Epidendroideae]]] based on the analysis of protein coding regions and rRNAs of orchid plastomes. This finding aligns with a previous study [5]. Three species of the genus *Gastrodia*, *G*. *elata*, *G*. *longyphyla*, and *G*. *pubilabiata*, formed a monophyletic group and the tribe Nervilieae was the sister group to Gastrodieae (Figure 5). In the case of Gastrodieae, Nervilieae, Diurideae, and Vanilleae, containing non-photosynthetic orchids, a long branch was formed, which might be due to reduced number of genes used in the analysis or elevated mutation rates in mycoheterotrophic species. A phylogenetic tree based on mitochondrial genes indicated that *G*. *pubilabiata* was part of a monophyletic group along with other Asparagales species (Figure 6). Unlike the plastome phylogenetic tree, non-photosynthetic species exhibited limited levels of long branches.

## 4. Materials and Methods

### 4.1. Plant Material and DNA Extraction

Plant samples were collected from Jeju Island, Republic of Korea. To protect plant habitat, only a few flowers were taken from individuals of *G. pubilabiata*. Fresh samples were ground into powder using liquid nitrogen in a mortar. Ground samples were then used to extract genomic DNA using a G-spin II Genomic DNA extraction Kit (Intron, Seoul, Republic of Korea) following the manufacturer’s manual. The quality of the extracted DNA was checked with a UV/VIS spectrophotometer. DNA was then stored in Plant DNA bank in Republic of Korea (PDBK2020-0086). Due to the limitations of sample collection, the voucher specimen was replaced with photographs of *G*. *pubilabiata*’s in its habitat. These photographs were deposited in Korea University Herbarium (KUS) under the voucher-specimen number of KUS2020-0086 (Appendix A). Overall analysis steps and procedures adopted in this study are summarized in Appendix A.

### 4.2. Sequencing, Assembly and Annotations

Extracted genomic DNA of *G. pubilabiata* was used for sequencing using two different NGS platforms: Illumina NovaSeq (Illumina, San Diego, CA, USA) and Pacbio Sequel (Pacbio, Menlo Park, CA, USA). A total of 224,926,562 NovaSeq reads and 746,492 Pacbio Sequel reads were produced. The quality of sequencing reads was examined using FastQC v.0.11.9 (https://www.bioinformatics.babraham.ac.uk/projects/fastqc, accessed on 29 June 2023) [71]. Raw reads from Illumina NovaSeq were about 50 gb. They were trimmed with BBduk 37.64, as implemented in Geneious 11.1.5 (length: 27 kmer) [72]. BBNorm 37.64 was used to normalize trimmed reads (target coverage level: 30; minimum depth: 12). Raw reads from Pacbio Sequel were about 5 gb. They were corrected and polished using trimmed NOVAseq reads with CANU v.1.9 [73].

Error corrected Pacbio reads were used for de novo assembly using FALCON v.0.3.0 [74,75] and CANU v.1.9. Mitogenome of *Allium cepa* (NC030100) and 18 mitochondrial sequences of *G. elata* (MF070084~MF070102) were used as references to verify putative mitochondrial contigs among de novo assembly results obtained using FALCON and CANU. Trimmed NovaSeq reads and selected contigs from FALCON and CANU results were used for hybrid de novo assembly using SPAdes v.3.14.1 [76]. Mitochondrial contigs were selected from SPAdes results by performing local BLASTN [77] and using BANDAGE v.0.8.0. [78]. The final selection of mitogenome was based on their coverage, which was obtained using hybridSPAdes as part of SPAdes v.3.14.1 [79]. The depth-coverage of mitogenomes of *G. pubilabiata* is represented in Appendix A.

Plastome sequences of *G. elata* (NC037409) and *Habenaria radiata* (NC035834) were used as references to identify plastome contigs from results of FALCON and CANU. Selected contigs were assembled using Geneious assembler in Geneious 11.1.5 to obtain complete plastome sequence. Trimmed NovaSeq reads were then mapped to the complete plastome sequence of *G*. *pubilabiata*’s using Geneious assembler, and the mapped NovaSeq reads were extracted. Extracted NovaSeq reads were used to perform de novo assembly using Geneious assembler to validate *G*. *pubilabiata*’s complete plastome sequence.

The complete plastome sequence of *G. pubilabiata* was annotated using BLASTN, tRNAscan-SE 2.0 [80], ORF finder, and the find annotation function in Geneious 11.1.5. Plastome sequences of *G*. *elata* and *H. radiata* were used as references. Alternative start codons (ACG and TTG) were also considered in ORF finder. Pseudogene judgement was performed based on the criteria described previously [43]. ORFs that were larger than 150 bp were translated into protein sequences. These sequences were used to perform a psi-blast search [81] using several reported Orchidaceae plastome sequences as references. A circular plastome map was generated using the OGdraw web server [82].

### 4.3. Phylogenetic Analysis

Ninety-two plastome sequences were downloaded from the NCBI for phylogenetic analysis (Appendix A), including 87 sequences for the Orchidaceae, four for the Asparagales, and one for Liliales. Seventy-nine protein coding genes and four rRNA genes were extracted from these plastome sequences. Each extracted gene was aligned using MAFFT v.7.450 [83]. Alignments were manually checked. All alignments were concatenated to a length of 85,749 bp and subjected to jModeltest in the CIPRES Science Gateway to determine the best fit model [84,85]. The best fit model was determined to GTR + I + G. Missing genes were treated as missing data. A maximum likelihood (ML) tree was constructed using RaxML-HPC2 on XSEDE in CIPRES Science Gateway with 100 bootstrap replicates [86]. The result ML tree was visualized using Treegraph 2.15.0-887 [87].

Thirty-eight mitogenome sequences were downloaded from the NCBI (Appendix A). Thirty-five coding genes and three rRNA genes were extracted from these mitogenomes. Each gene was aligned using MAFFT v.7.450. All alignments were manually checked. Alignments were concatenated to a total length of 51,862 bp and then subjected to jModeltest in CIPRES Science Gateway to determine the best model. An ML tree was constructed using RaxML-HPC2 on XSEDE in CIPRES Science Gateway with a GTR + I + G model and 100 bootstrap replicates. The obtained ML tree was graphically represented using Treegraph 2.15.0-887.

Eighteen mitochondrial sequences of *G. elata* (MF070084~MF070102) and two plastome sequences of *G. elata* (NC037409) and *G. longistyla* (MW879162) were used to compare organelles among the genus *Gastrodia*. Simple sequence repeats (SSRs) were identified using Phobos 3.3.12 implemented in Geneious 11.1.5 [88]. REPuter (https://bibiserv.cebitec.uni-bielefeld.de/reputer, accessed on 28 June 2023) was used to distinguish palindromic repeats among organelles with default options [89]. The repeat finder in Geneious 11.1.5 was used to find repeats larger than 50 bp with a perfect match option. Repeat information was summarized and presented graphically using Circos v.0.69-9 [90].

A total of 5199 reported angiosperm plastome sequences and 219 mitogenome sequences were downloaded from the NCBI and used as a local BLAST database. Plastome and mitogenome sequences of *G. pubilabiata* were divided into 150 bp fragments, which were then used to perform local BLASTN to identify potential instances of IGT or HGT. Results were summarized in terms of chloroplast counts and mitochondria counts based on the number of results generated. Summarized data were visualized using ggplot2 in R [91]. Mitogenome regions skewed toward to plastome in BLASTN results were carefully evaluated, and also plastome regions skewed toward to mitogenome were evaluated (Appendix A). Based on these data, mitochondrial plastome (MTPT) sequences were extracted to construct a phylogenetic tree. Fifty-three plastome sequences of the main angiosperm lineage were used as references for this phylogenetic tree. For *psa*A and *rpo*B genes, a total of 301 Malpighiales and Celastrales plastome sequences were used as references to construct the ML tree. All obtained ML trees were visualized using Treegraph2. In order to estimate the HGT time, the divergence time of the *Gastodea* lineage were adopted from the previous study [43].

## 5. Conclusions

In this research we decode the entire plastome and mitogenome from a non-photosynthetic and mycoheterotrophic orchid species, *G. pubilabiata*. As expected, the plastome is dramatically reduced in size to be only 30,698 bp. This reduction has led to the loss of most photosynthesis-related genes and some housekeeping genes in the plastome, which now only contains 19 protein coding genes, three tRNAs, and three rRNAs. Therefore, the chloroplast lost its function and the genome faced strong relaxed selection pressure. In contrast, the mitogenome of *G*. *pubilabiata* is 867,349 bp in length and contained 38 protein coding genes, nine tRNAs, and three rRNAs. The gene content of the *G. pubilabiata* mitogenome is similar to the typical plant mitogenomes, even though the genome size is twice larger than the typical ones. Plastid rRNA fragments were found at a high frequency in the mitogenome. In addition, seven plastid protein coding gene fragments (*ndh*C, *ndh*J, *ndh*K, *psa*A, *psb*F, *rpo*B, and *rps*4) were also identified in the mitogenome of *G*. *pubilabiata*. This horizontal gene transfer (HGT) from plastome to mitogenome occurred before losses of photosynthesis related genes from the chloroplast of *G*. *pubilabiata*. The HGT have occurred approximately 30 million years ago. This is the first documentation of the HGT between two cellular organelles of the orchid family. Furthermore, this is the first report for the HGT time from the orchid family. This finding will expand our knowledge for understanding of the orchid genome evolution. We expect more similar results from the orchid family which contains a wide range of nonphotosynthetic species.

## Figures and Tables

**Figure 1 ijms-24-11448-f001:**
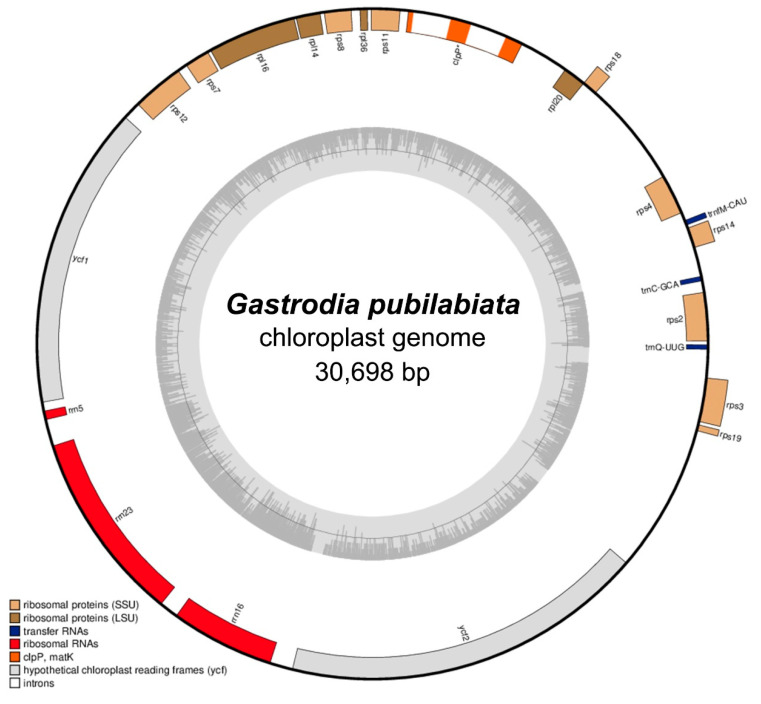
The complete plastome of *G. pubilabiata*. The plastome is 30.698 bp in length and lack IR region. It contains 19 protein coding genes, three tRNA genes, and three rRNA genes.

**Figure 2 ijms-24-11448-f002:**
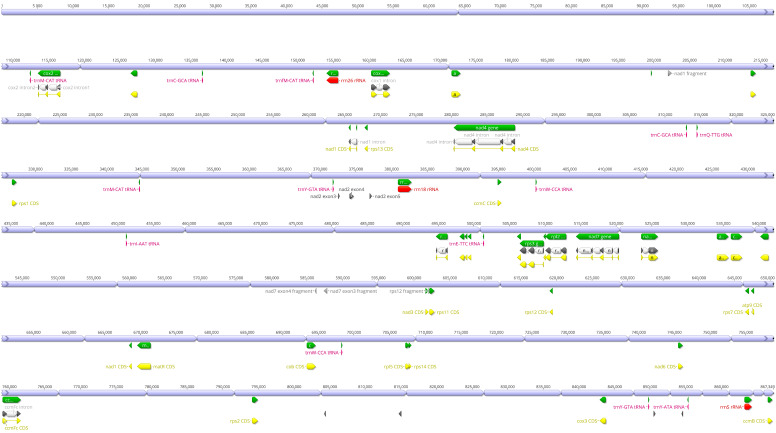
The complete mitogenome of *G. pubilabiata*. It is composed of 44 mitochondrial contigs. A total length of mitogenome is 867,349 bp in length and the genome include 38 protein coding genes, nine tRNA genes, and three rRNA genes.

**Figure 3 ijms-24-11448-f003:**
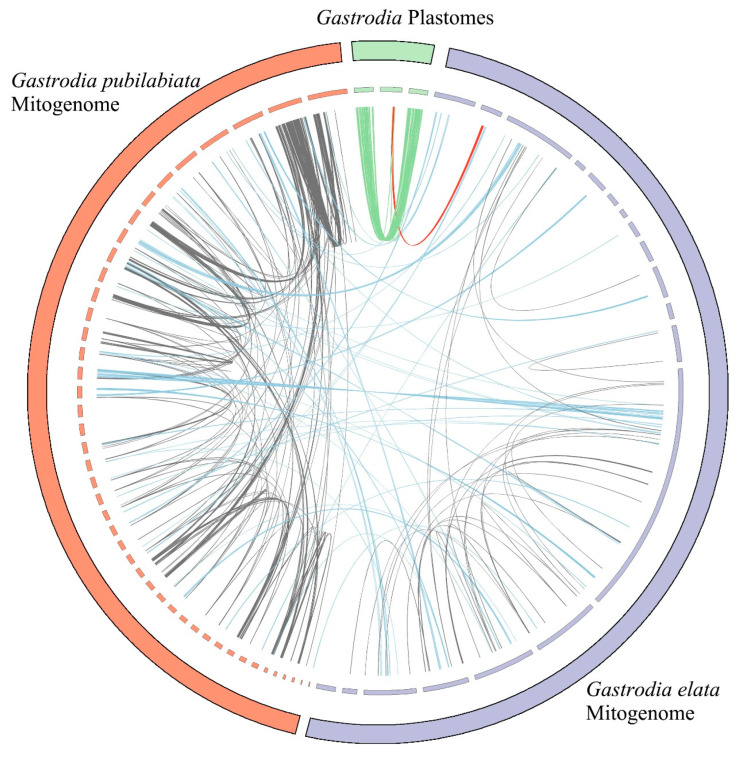
Results of repeat analysis between *Gastrodia* organelle genomes. The blue lines represent repeats between mitogenomes of different species, the green lines represent repeats between plastomes of different species, the grey lines indicate repeats between mitogenomes of the same species, and the red lines indicate repeats between plastomes and mitogenomes, respectively.

**Figure 4 ijms-24-11448-f004:**
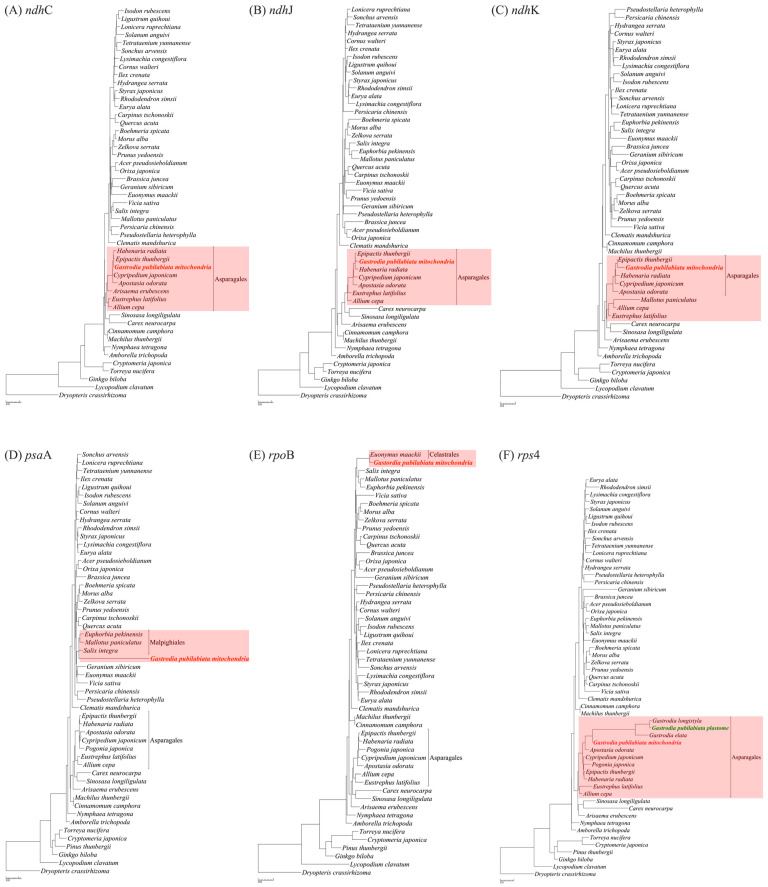
A maximum likelihood (ML) phylogenetic tree of six regions found to be plastid-origin coding regions in the mitogenome (*ndh*C, *ndh*J, *ndh*K, *psa*A, *rpo*B, and *rps*4). The position of *G. pubilabiata* based on mitogenome is highlighted in red. The position of *G*. *pubilabiata* based on plastome is highlighted in green.

**Figure 5 ijms-24-11448-f005:**
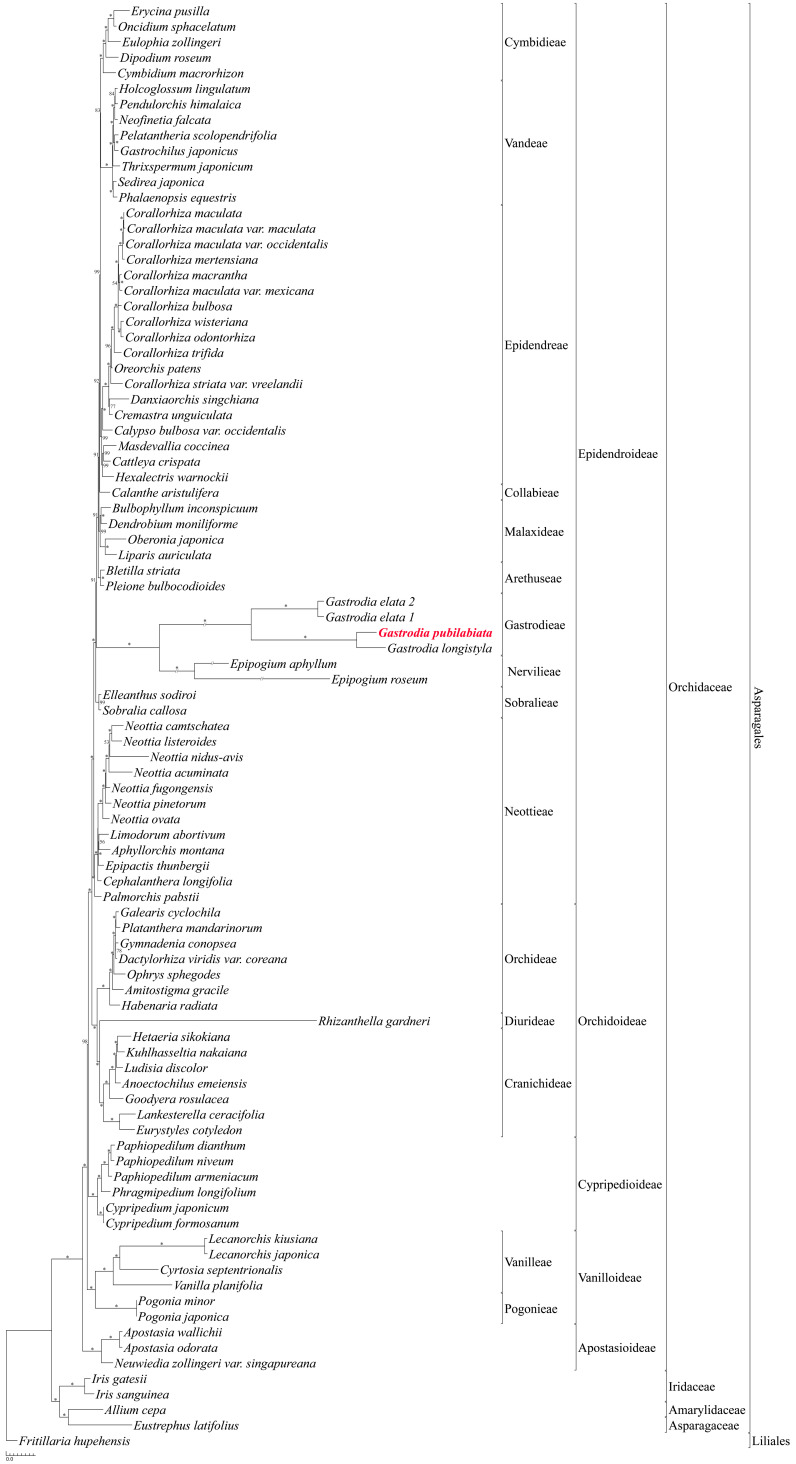
A maximum likelihood (ML) phylogenetic tree inferred from 92 species (87 species from Orchidaceae, four species from Asparagales, and one species from Liliales were used). The nucleotide sequences of 79 protein coding genes and four rRNA genes were aligned independently and concatenated to be a single data matrix. The alignment was 85,749 bp and the tree was constructed by RaxML with the GTR + I + G model (−621,773.060310 of ML value). The scientific name of *G. pubilabiata* is highlighted with red color. The numbers above or below the nodes are the bootstrap values. Asterisk (*) mark above or below the nodes indicates the bootstrap value of 100.

**Figure 6 ijms-24-11448-f006:**
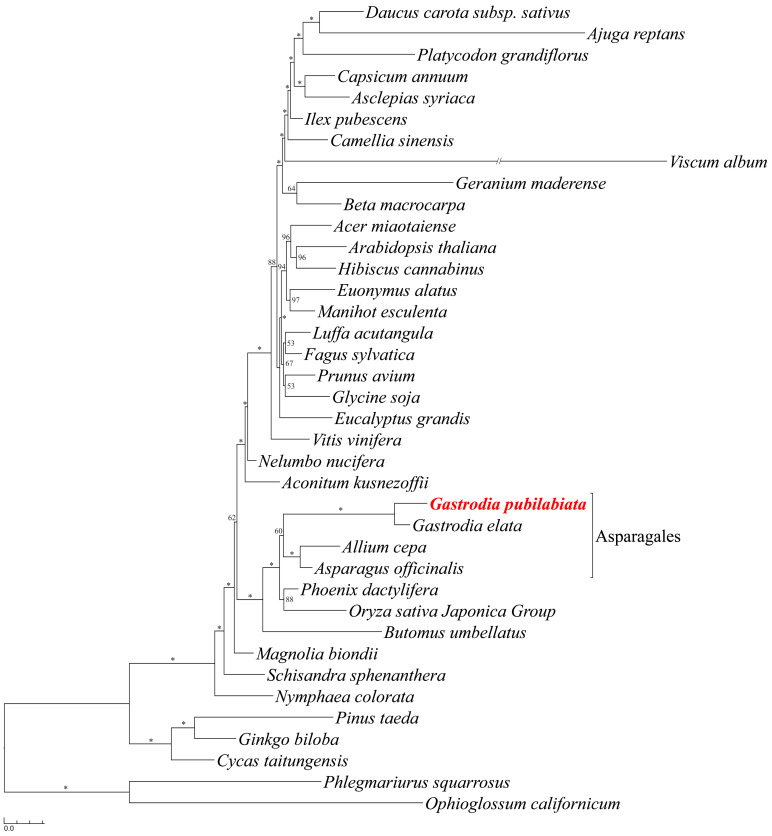
A maximum likelihood (ML) phylogenetic tree inferred from 38 species. The nucleotide sequences of 35 protein coding genes and three rRNA genes were aligned independently and concatenated to be a single data matrix. The alignment was 51,862 bp and the tree was constructed by RaxML with the GTR + I + G model (−318,584.844612 of ML value). The scientific name of *G. pubilabiata* is highlighted with red color. The numbers above or below the nodes are the bootstrap values and the asterisk (*) mark above or below the nodes indicate the bootstrap value of 100.

**Figure 7 ijms-24-11448-f007:**
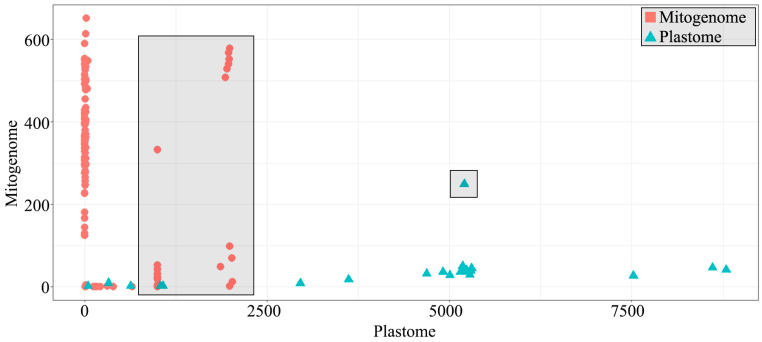
A dot-plot based on local BLASTN search results. Blue triangular dots represent plastome-origin sequences. Red circular dots indicate mitogenome-origin sequences. Irregular search results are highlighted with gray boxes.

**Table 1 ijms-24-11448-t001:** The general features of *G. pubilabiata*’s plastome and mitogenome.

	Plastome	Mitogenome
Contigs	1	44
Total Length (bp)	30,698	867,349
GC (%)	24.9	42.1
Depth Coverage (×)	236.0	150.0
Protein Coding Genes	19	38
tRNAs	3	9
rRNAs	3	3

**Table 2 ijms-24-11448-t002:** The general features of reported mitogenomes of monocotyledon.

Scientific Name	Accesions	Contigs	Total Length (bp)	Protein Coding Genes	tRNAs	rRNAs
*Gastrodia pubilabiata*	OR004100–OR004143	44	867,349	38	9	3
*Gastrodia elata*	MF070084~ MF070102	19	1,339,825	38	14	3
*Allium cepa*	NC030100	1	316,363	24	5	3
*Asparagus officinalis*	NC053642	1	492,062	36	14	3
*Butomus umbellatus*	NC021399	1	450,826	28	12	4
*Oryza sativa*	NC011033	1	490,520	31	15	3
*Phoenix dactylifera*	NC016740	1	715,001	37	16	3

## Data Availability

The newly sequenced plastome and mitogenome of *G. pubilabiata* were submitted to GenBank and the accession number are OR004100-OR004143 for mitogenome and OR031839 for plastome, respectively.

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
