# Peer review of "Ancient Horizontal Gene Transfers from Plastome to Mitogenome of a Nonphotosynthetic Orchid, Gastrodia pubilabiata (Epidendroideae, Orchidaceae)"

_ijms, 2023, doi:10.3390/ijms241411448_

Round 1
Reviewer 1 Report
The manuscript titled “Ancient horizontal gene transfers from plastome to mitogenome of a nonphotosynthetic orchid, Gastrodia pubilabiata (Epidendroideae, Orchidaceae)”. The study performed by the authors decoded the entire mitogenome G. pubilabiata, consisting of 44 contigs, and found high frequency of plastid rRNA fragments (ndhC, ndhJ, ndhK, psaA, psbF, rpoB, and rps4). Phylogenetic trees using these fragments suggested horizontal gene transfer (HGT) occurred before photosynthesis-related gene losses, leading to the lineage of G. pubilabiata. HGT is estimated to have occurred approximately 30 million years ago. Further, the findings of this investigation will advance researchers understanding of the mitogenome's processes of evolution in non-photosynthetic orchid species. Manuscript scientifically contains all the necessary elements.
However, the manuscript still needs some minor improvisation. Therefore, I recommend the authors to incorporate the following points in the manuscript for further consideration.
Plant scientific names should be in full form in first mention rest should be abbreviated. Authors should revise this throughout the manuscript. Eg. Line 106 - Gastrodia pubilabiata and Line 134, 137should be G. pubilabiata.
Authors are advised to provide an overall framework figure listing all the analysis and steps done in this paper in materials and methods.
Authors must concentrate on the formatting and use of symbols, etc., throughout the manuscript. e.g., on line 186, the coding regions of the mitogenome might be in italics.
Enhance the figure 2 resolution.
Figure S2 is not clear. Enhance and revise it.
The materials and methods section looks shallow. Improve it with more detailed analysis.
Authors should provide a separate conclusion section and write a few lines about future perspectives or hypotheses about the study. It will be useful to the readers for ease of understanding.
Author Response
Dear Reviewer 1:
I greatly appreciate you for valuable comments. I adopted all of your comments in this revised version.
Response to reviewer 1
1) Plant scientific names should be in full form in first mention rest should be abbreviated. Authors should revise this throughout the manuscript. Eg. Line 106 - Gastrodia pubilabiata and Line 134, 137should be G. pubilabiata.
Reply) Yes. I changed all scientific names to abbreviated forms after the first appearance.
2) Authors are advised to provide an overall framework figure listing all the analysis and steps done in this paper in materials and methods.
Reply) Yes. I added a supplemental figure S5 outlined the analysis steps and procedures rather than a main figure. (Lines 402-404, 495-496)
3) Authors must concentrate on the formatting and use of symbols, etc., throughout the manuscript. e.g., on line 186, the coding regions of the mitogenome might be in italics.
Reply) Yes. I corrected all the gene names in italic. (Lines 189)
4) Enhance the figure 2 resolution.
Reply) Yes. I replaced the figure 2 with a higher resolution and larger one.
5) Figure S2 is not clear. Enhance and revise it.
Reply) Yes. I replaced the figure S2 with a higher resolution and larger one.
6) The materials and methods section looks shallow. Improve it with more detailed analysis.
Reply) Yes. I added three more sentences. The addition of outline figure S5 will improve this section. (Lines 474-476, and 481-482)
7) Authors should provide a separate conclusion section and write a few lines about future perspectives or hypotheses about the study. It will be useful to the readers for ease of understanding.
Reply) Yes. I added a new conclusion paragraph (Lines 372-390).
I believe this revision significantly improve the manuscript quality. I also appreciated to you for the fast manuscript handling.
Sincerely,
Ki-Joong Kim, Prof.
Division of Life Sciences
Korea University
Seoul 02841, Korea
Reviewer 2 Report
The manuscript “Ancient horizontal gene transfers from plastome to mitogenome of a nonphotosynthetic orchid, Gastrodia pubilabiata (Epidendroideae, Orchidaceae)” determine possible gene transfer events between the plastome and the mitogenome, individual BLASTN searches, using all available orchid plastome sequences and flowering plant mitogenome sequences.
The manuscript is prepared professionally. It includes a well-crafted abstract and an exhaustive introduction that justifies the research undertaken. The introduction points to the deficiencies in the literature on the subject. The aim is clearly defined. Modern analytical methods were used in the research. The discussion of the results is well prepared. The conclusions are well-defined. The illustrative material is appropriate.
In my opinion, the manuscript after corrections, will be suitable for publication in a journal.
Detailed comments:
Abstract: Should include some more numeric data obtained from the study
Introduction - The introduction is enough in my opinion. Introduction needs some minor changes
Line 30-32 The advancement in sequencing technology has led to a surge in research on nucleotide sequences of flowering plants and their use for evolutionary and phylogenetic studies. This sentence needs references. I suggest below ones.
Zia A.; Zahoor M.; Abdullah A.; Nisar A.; Batool N.; Bibi A.; Saba K.; Ahmed I.; Duran ST.; Ipek A. Identification of SNP markers linked to Rf locus in carrot using GBS. Turk. J. Agric. For. 2022, 46 (6): 898-907. https://doi.org/10.55730/1300-011X.3051.
Karatas A.; Gurel E.; Wahed MT. De novo assembly and characterisation of chloroplast genomes of broccoli cvs. Marathon and Green sprout using next generation sequencing. Turk. J. Agric. For. 2022, 46 (4): 523-535. https://doi.org/10.55730/1300-011X.3023.
Is it possible to make better Figure 2?
Any novelty in terms of scientific idea and experimental design? Explain more
What is the hypothesis of the study? It should be stated in the aim of the study
The manuscript “Ancient horizontal gene transfers from plastome to mitogenome of a nonphotosynthetic orchid, Gastrodia pubilabiata (Epidendroideae, Orchidaceae)” determine possible gene transfer events between the plastome and the mitogenome, individual BLASTN searches, using all available orchid plastome sequences and flowering plant mitogenome sequences.
The manuscript is prepared professionally. It includes a well-crafted abstract and an exhaustive introduction that justifies the research undertaken. The introduction points to the deficiencies in the literature on the subject. The aim is clearly defined. Modern analytical methods were used in the research. The discussion of the results is well prepared. The conclusions are well-defined. The illustrative material is appropriate.
In my opinion, the manuscript after corrections, will be suitable for publication in a journal.
Detailed comments:
Abstract: Should include some more numeric data obtained from the study
Introduction - The introduction is enough in my opinion. Introduction needs some minor changes
Line 30-32 The advancement in sequencing technology has led to a surge in research on nucleotide sequences of flowering plants and their use for evolutionary and phylogenetic studies. This sentence needs references. I suggest below ones.
Zia A.; Zahoor M.; Abdullah A.; Nisar A.; Batool N.; Bibi A.; Saba K.; Ahmed I.; Duran ST.; Ipek A. Identification of SNP markers linked to Rf locus in carrot using GBS. Turk. J. Agric. For. 2022, 46 (6): 898-907. https://doi.org/10.55730/1300-011X.3051.
Karatas A.; Gurel E.; Wahed MT. De novo assembly and characterisation of chloroplast genomes of broccoli cvs. Marathon and Green sprout using next generation sequencing. Turk. J. Agric. For. 2022, 46 (4): 523-535. https://doi.org/10.55730/1300-011X.3023.
Is it possible to make better Figure 2?
Novelty in terms of scientific idea and experimental design? Explain more
Hypothesis of the study? It should be stated in the aim of the study more
Author Response
Dear Reviewer 2:
I greatly appreciate to you for your valuable comments. I adopted all of your comments in this revised version.
Response to reviewer 2
1) Abstract: Should include some more numeric data obtained from the study
Reply) Yes. I added two new sentences with numeric data (Lines 15-16, 19-20).
2) Introduction - The introduction is enough in my opinion. Introduction needs some minor changes
Line 30-32 The advancement in sequencing technology has led to a surge in research on nucleotide sequences of flowering plants and their use for evolutionary and phylogenetic studies. This sentence needs references. I suggest below ones.
Zia A.; Zahoor M.; Abdullah A.; Nisar A.; Batool N.; Bibi A.; Saba K.; Ahmed I.; Duran ST.; Ipek A. Identification of SNP markers linked to Rf locus in carrot using GBS. Turk. J. Agric. For. 2022, 46 (6): 898-907. https://doi.org/10.55730/1300-011X.3051.
Karatas A.; Gurel E.; Wahed MT. De novo assembly and characterisation of chloroplast genomes of broccoli cvs. Marathon and Green sprout using next generation sequencing. Turk. J. Agric. For. 2022, 46 (4): 523-535. https://doi.org/10.55730/1300-011X.3023.
Reply) Yes. I added two new references 1 and 2. But, I added more appropriate references rather than the above suggested ones. (Lines 523-527).
3) Is it possible to make better Figure 2?
Reply) Yes. I replaced the figure 2 with a higher resolution and larger one.
4) Any novelty in terms of scientific idea and experimental design? Explain more
Reply) Yes. I added a new conclusion paragraph (Lines 372-390). In this paragraph, I state that this is the first report of HGT in the orchid family.
5) What is the hypothesis of the study? It should be stated in the aim of the study
Reply) Yes. I modified the last sentence of introduction in order to clarify the hypothesis and aim of this study (Lines 102-106).
I believe this revision significantly improves the manuscript quality. I also appreciated the reviewers for fast manuscript handling.
I believe this revision significantly improve the manuscript quality. I also appreciated to you for the fast manuscript handling.
Sincerely,
Ki-Joong Kim, Prof.
Division of Life Sciences
Korea University
Seoul 02841, Korea